# In Vitro Antioxidant Activity of *Litsea martabanica* Root Extract and Its Hepatoprotective Effect on Chlorpyrifos-Induced Toxicity in Rats

**DOI:** 10.3390/molecules26071906

**Published:** 2021-03-28

**Authors:** Phraepakaporn Kunnaja, Sunee Chansakaow, Absorn Wittayapraparat, Pedcharada Yusuk, Seewaboon Sireeratawong

**Affiliations:** 1Department of Medical Technology, Faculty of Associated Medical Sciences, Chiang Mai University, Chiang Mai 50200, Thailand; phraepakaporn.k@cmu.ac.th; 2Department of Pharmaceutical Sciences, Faculty of Pharmacy, Chiang Mai University, Chiang Mai 50200, Thailand; sunee.c@cmu.ac.th; 3Highland Research and Development Institute (Public Organization), Chiang Mai 50200, Thailand; absornw@hrdi.or.th (A.W.); npedcharada@hrdi.or.th (P.Y.); 4Department of Pharmacology, Faculty of Medicine, Chiang Mai University, Chiang Mai 50200, Thailand

**Keywords:** *L. martabanica*, antioxidant, anti-pesticide, medicinal plant, acetylcholinesterase activity, hepatoprotective

## Abstract

In Thailand, people in the highland communities whose occupational exposure to pesticides used the root of *Litsea martabanica* as a detoxifying agent. However, the scientific data to support the traditional use of this plant are insufficient. This study aimed to evaluate the antioxidant activity and anti-pesticide potential of *L. martabanica* root extract. Antioxidant properties were investigated by 2,2′-diphenyl-1-picrylhydrazyl (DPPH) assay, superoxide radicals scavenging assay, 2,2′-azino-bis(3-ethylbenzothiazoline-6-sulfonic acid) (ABTS) assay, ferric reducing antioxidant power (FRAP), and total phenolic content determination. In all assays, *L. martabanica* extracts and their fractions exhibited high antioxidant activities differently. The water extract is traditionally used as a detoxifying agent. Therefore, it was chosen for in vivo experiments. The rats received the extract in a way that mimics the traditional methods of tribal communities followed by chlorpyrifos for 16 days. The results showed that acetylcholinesterase activity decreases in pesticide-exposed rats. Treatment with the extract caused increasing acetylcholinesterase activity in the rats. Therefore, *L. martabanica* extract may potentially be used as a detoxifying agent, especially for the chlorpyrifos pesticide. The antioxidant properties of *L. martabanica* may provide a beneficial effect by protecting liver cells from damage caused by free radicals. Histopathology results revealed no liver cell necrosis and showed the regeneration of liver cells in the treatment group. *L. martabanica* extract did not cause changes in behavior, liver weight, hematological and biochemical profiles of the rats.

## 1. Introduction

*Litsea* is evergreen or a rare deciduous, dioecious tree or shrub in the family Lauraceae. There are over 400 species, mainly in tropical and subtropical Asia, but with a few species in the islands of the Pacific, Australia, and North and Central America, and 27 species in Thailand [1,2]. The plant in *Litsea* species has been used as traditional herbal medicines for thousands of years [3]. Twenty plants of the genus *Litsea* are found to be important traditional medicines in China for treating diarrhea, stomachache, dyspepsia, gastroenteritis, diabetes, edema, cold, arthritis, asthma, pain, traumatic injury, etc. [4]. One of the *Litsea* plants, *Litsea cubeba,* different parts of this plant, such as bark, leaf, root, and fruits, are used for treating many kinds of diseases [2]. This plant also exhibited antimicrobial, antioxidant, anti-cancer, anti-inflammatory, anti-diabetic, anti-insecticidal, and hepatoprotective activities [2].

*Litsea martabanica* (Kurz) Hook.f. is one of the species found in Thailand and is also distributed in China and Myanmar (Figure 1) [1]. The history of utilization of this plant is based on the wisdom of the highland communities. Various parts, i.e., the roots, leaves, and stems, have been traditionally used as medicine in the highland area of the northern part of Thailand for curing kidney disease, curing toxic allergy symptoms, and detoxification [5]. Detoxification or removal of toxins in humankind is an alternative way to promote good health for the people in the highland area who use pesticides and insecticides in daily life. The accumulation of pesticides in the body may be through the consumption of contaminated food or exposure in the occupational environment [6]. Organophosphate and carbamate are commonly used pesticides due to a short half-life and are non-persistent in the environment [7]. These pesticides cause acetylcholinesterase (AChE) enzyme inhibition, leading to an increase of acetylcholine (ACh) at the synapses and neuromuscular junctions. Organophosphates (OPs) are irreversible AChE inhibitors [8]. AChE inhibition causes muscarinic and nicotinic toxicity, including cramps, increased salivation, lacrimation, muscular weakness, paralysis, muscular fasciculation, diarrhea, and blurred vision [9]. It has been reported that OP pesticides induced reactive oxygen species (ROS) generation, which alters an antioxidant system leading oxidative damage to the cells [10]. Humans with occupational exposes to the pesticide have been reported to have increased lipid peroxidation and significantly reduced AChE activity [11,12]. The levels of antioxidant enzymes, catalase, superoxide dismutase, glutathione peroxidase, and glutathione reductase, as well as non-enzymatic antioxidant, reduced glutathione, were changed in organophosphate poisoning [13,14]. Prolonged exposure to these pesticides was reported to be associated with various types of cancers [15] and pathologic liver diseases such as hepatitis, fibrosis, and cirrhosis [16,17].

The liver is the first organ to eliminate potentially harmful xenobiotics and pesticides through the cytochrome P450 enzyme metabolism. However, a high dose of pesticide exposure may reduce the detoxifying function of the liver, leading to liver cell injury. The pesticide-induced liver toxicity involves ROS generation, which causes oxidative damage to the liver cells. Some reports suggested that antioxidant substances may be beneficial in the treatment of acute organophosphate pesticide poisoning [18,19]. The generation of oxygen-free radicals during pesticide exposure can be neutralized by various enzymatic and non-enzymatic antioxidant systems [20,21]. Plants are considered as an important source to meliorate ROS. The non-enzymatic antioxidants, including ascorbic acid, glutathione, proline, carotenoids, phenolic acids, flavonoids, tannins, etc., were found in many plants [22]. Some research findings purposed the potential use of many non-enzymatic antioxidant substances to eliminate the adverse effect of pesticides [23]. An in vitro experimental study has reported that vitamin C and vitamin E may ameliorate oxidative stress induced by organophosphate pesticides through the decreasing of lipid peroxidation in erythrocytes [24,25]. A study in Wistar rats has demonstrated that vitamin C pretreatment improves sensorimotor and cognitive functions in acute short-term chlorpyrifos-exposed rats [26].

Besides indigenous knowledge, there are no scientific data, especially pharmacological activities, to support the traditional use of *L. martabanica* as a detoxifying agent. In the present study, we investigated the anti-pesticide potential of *L. martabanica* extract on rats. AChE activity, which represents pesticide exposure, was measured in the rats’ blood samples. Other parameters, including body weight change, blood hematology, blood chemistry profiles, and internal organ weight, were determined. Many plants in the *Litsea* species have been reported to possess high antioxidant activity [2,3]. Therefore, we investigated the in vitro antioxidant activity of *L. martabanica* extract as well.

## 2. Results

### 2.1. Microscopic Character and Chemical Pattern of the Extract of L. martabanica

The raw material of *L. martabanica* (root) was extracted by decoction, following indigenous methods using water as a solvent. The extract was filtrated, concentrated until % Brix = 3, added pharmaceutical aids adsorbent (Cab-O-Sil^®^) as a carrier, and then dried by a spray dryer. The obtained brownish powder was 5.38% *w/w*. Microscopic characteristics were analyzed by microscope connect with camera-lucida (Figure 2). Starch grains, fibers, epidermis, fibers containing starch grains, bordered vessels, and stone cells are major tissues found in the powder of the root of *L. martabanica*. The microscopic characteristics can be used for identification because the identity tissue is revealed in each different plant. The monograph of the root of *L. martabanica* has not been officially found in any pharmacopeia or textbooks. As a result, the specification of the root of this plant can be used as a reference for the quality control of raw materials in a further study.

Phytochemical screening was carried out for the root of *L. martabanica*, as described in the standard method [27]. The phenolics, flavonoids, and terpenes were found in the test sample. The results of the evaluation of their quality following the methods described in the Thai herbal pharmacopeia 2018 [27] are shown in Table 1.

The chemical profile was performed with the suitable mobile phase, compared with known bioactive compounds, and visualized under UV 254 nm and 366 nm. The densitogram of the water extract and ethanolic extract indicated 3 and 7 major bands, respectively (Figure 3A,B and Figure 4). The ethanolic extract and water extract of the root of *L. martabanica* did not consist of apigenin, caffeic acid, gallic acid, kaemferol, pinene, and quercetin compared to the standard rate of flow (Rf) value. The fractions isolated by the different solvent polarities found that the chloroform fraction and aqueous alcoholic fraction revealed the same component at Rf 0.17, 0.25, and 0.38. Therefore, it may be that the bioactive compound(s) found in this plant is partly used. The phytochemical components of this plant have not been reported. Therefore, bioassay-guided isolation will need to be studied further to find bioactive compounds. In this study, the high-performance thin-layer chromatography or HPTLC chromatogram did not indicate the chemical components, but fingerprinting can be used to analyze the quantification of phytochemical herbal products, and examine adulteration in herbal formulations.

### 2.2. 2,2′-diphenyl-1-picrylhydrazyl (DPPH) Scavenging Activity

*L. martabanica* fractions inhibited DPPH free radicals in a concentration-dependent manner, as shown in Figure 5. The reference standard gallic acid had the highest efficacy in scavenged DPPH free radicals. The result of the half maximal inhibitory concentration (IC_50_) values is shown in Table 2. Among these fractions, aqueous ethanol fraction, crude water extract, and crude ethanol extract exhibited higher antioxidant properties than chloroform and hexane fraction.

### 2.3. Superoxide Radical Scavenging Activity

In this assay, the extracts and fractions of *L. martabanica* inhibited superoxide radical generation in a concentration-dependent manner (Figure 6). Free radical scavenging efficacy of different test samples and gallic acid was in the following order: gallic acid > aqueous ethanol fraction > crude water extract > crude ethanol extract > CHCl_3_ fraction > hexane fraction.

### 2.4. 2,2′-Azino-Bis-(3-Ethylbenzothiazoline-6-Sulfonic Acid) (ABTS) Radical Scavenging Activity

The ability of an extract to scavenge the ABTS radical is shown in Table 3. The crude ethanol extract possesses the most superior ability to scavenge free radicals as compared to other fractions. Radical scavenging efficacy in this assay was in the following order: crude ethanol extract > CHCl_3_ fraction > hexane fraction > crude water extract > aqueous ethanol fraction.

### 2.5. Ferric Reducing Antioxidant Power (FRAP)

As shown in Table 3, the CHCl_3_ fraction and crude ethanol extract of *L. martabanica* showed a greater FRAP value than other test samples. The FRAP values were 1554.1 mM and 1376.2 mM Fe (II)/g extract, respectively. The other test samples, aqueous ethanol fraction, crude water extract, and hexane fraction, showed FRAP values of 418.6 mM, 368.9 mM, and 275.4 mM Fe (II)/g extract, respectively.

### 2.6. Total Phenolic Content (TPC)

The amount of total phenolics varied widely in *L. martabanica* extracts and their fractions from 39.3 to 173.1 mg gallic acid equivalent (GAE)/g dry extract (Table 3). Among the test samples, the CHCl_3_ fraction displayed the highest amount of total phenolics. The phenolic content was found to be in the order of the CHCl_3_ fraction > crude ethanol extract > hexane fraction> crude water extract > aqueous ethanol fraction.

The observed results from antioxidant assays demonstrated that chloroform fraction possesses high antioxidant property over the other fraction, followed by crude ethanol extract, aqueous ethanol fraction, water extract, and hexane fraction, respectively. However, the crude water extract is traditionally used as a detoxifying agent by local people in the highland communities. Therefore, the water extract of *L. martabanica* was typically chosen for in vivo experiments.

### 2.7. Anti-Pesticide Potential

The result of the AChE activity is shown in Figure 7. AChE activity of the control rats received chlorpyrifos significantly decrease from the normal rats. However, co-treatment of *L. martabanica* and chlorpyrifos tended to increase AChE activity. The high efficacy of the extract in increasing AChE activity was seen in the high doses of the extract at 750 and 250 mg/kg. AChE activity significantly increased when compared with control rats. Thus, this extract may have the potential to reverse organophosphate poisoning.

### 2.8. Observation of Behavioral Change and Toxicological Signs

The rats in the control group showed signs of toxicity such as piloerection, irregular respiratory patterns, and isolation from the group. *L. martabanica* extract treatment rats did not show any signs of toxicity and revealed behavior similar to the rats in the normal groups.

### 2.9. Body Weight Change, Organs Weight, and Histological Examination Results

On day 16, the bodyweight of the control rats and *L. martabanica*-treated rats significantly decreased compared with the normal rats (Figure 8). Gross pathological examination revealed normal internal organs in all rats. The liver weight of extract-treated rats did not differ from control rats, except for the liver weight of rats that received the *L. martabanica* extract at doses of 75 and 25 mg/kg (Table 4). However, the liver weight of all extract treatment groups did not differ from the normal rats.

The results of liver pathology are shown in Figure 9. In the control group, the sinusoids are dilated or widen in comparison to the normal group. The scattered foci of hepatic necrosis in zone 2 were observed. No centrilobular necrosis is usually associated with congestion in this group. The liver histology of the *L. martabanica*-treated group is not comparable to the normal group, since the sinusoids are widened. However, no hepatic necrosis is noted. The cells are varied in shapes and sizes but their nuclei are vesicles with small nuclei.

### 2.10. Hematology Analysis

As shown in Table 5, the hematological parameters such as mean corpuscular volume (MCV), mean corpuscular hemoglobin (MCH), mean corpuscular hemoglobin concentration (MCHC), and platelet (PLT) significantly increased in the control rats. Red blood cell (RBC), hemoglobin (HBG), and hematocrit (HCT) tended to decrease, but no statistical difference was observed. *L. martabanica* extract treatment could alter RBC, HBG, HCT, MCV, MCH, MCHC, and PLT parameters close to the normal rats. Neutrophils and lymphocytes significantly decreased at high doses of treatment. However, at medium doses, these values did not differ from the normal rats. These effects may result from the biological variation among the rats. All hematological parameters were in the normal reference range for rats [28].

### 2.11. Blood Chemistry Analysis

The results of blood chemistry profiles are presented in Table 6. The markers of kidney function blood urea nitrogen (BUN) and creatinine (Cr) levels tended to increase in the control group, but did not show a statistically significant difference. The rats treated with *L. martabanica* at doses of 7.5 and 2.5 mg/kg showed significantly decreased BUN and Cr levels compared with control rats. The blood chemistry values total protein (TP), albumin (ALB), total bilirubin (TB), direct bilirubin (DB), aspartate aminotransferase (AST), alanine aminotransferase (ALT), and alkaline phosphatase (ALP) typically represent liver function. These values significantly increased in control rats. However, *L. martabanica* treatment could shift abnormal parameters close to the normal rats [28].

## 3. Discussion

The root of *L. martabanica* was traditionally used for detoxification by the highland communities in Thailand. However, scientific data to support the traditional use of this plant are still insufficient. In the present study, we investigated the microscopic character of *L. martabanica*, as well as the chemical patterns, antioxidant activity, and anti-pesticide potential of *L. martabanica* root extract.

Antioxidants are essential substances to scavenge free radicals and prevent oxidative damage to the cells. The single antioxidant method is insufficient to study the antioxidant capacity of the plant samples. The measurement of antioxidant activity needs to use the various models to evaluate antioxidant mechanisms [29]. DPPH assay, superoxide radical assay, ABTS assay, FRAP assay, and total phenolic content are the methods used for screening the antioxidant activity of plant samples [30,31]. The results of all antioxidant assays indicated that *L. martabanica* root extract exhibited high antioxidant activities. Various plants in the genus Litsea provided a rich source of natural antioxidants to possess free radical scavenging potential [3]. *L. martabanica* extracts and their fractions demonstrated different antioxidant efficacy in each assay model. These results correlated with the previous scientific reports. For example, *L. coreana* var. lanuginose or Hawk primary leaf tea infusion (HPI) had high polyphenols contents and exhibited scavenging activity in DPPH and FRAP assay [32]. The methanol extract of *L. cubeba* showed remarkable antioxidant activity in DPPH assay, peroxidase/guaiacol assay, and thiobarbituric acid (TBA) test in comparison with α-tocopherol and ascorbic acid [33]. The study of chemical constituents of 20 *litsea* plant species in China has been reported compose of flavonoids, terpenoids, alkaloids, butanolides and butenolactones, lignans, amides, steroids, fatty acids, and megastigmanes [4,34].

Flavonoids and terpenoids are important bioactive constituents in this genus and exert a therapeutic effect in preventing or slowing oxidative stress-related diseases [4]. Phenolic compounds are considered secondary metabolites synthesized by plants. These compounds play an essential role in multiple biological effects, including antioxidant activity by scavenging free radicals [35,36,37,38]. The results of chemical composition in this study showed the presence of phenolics, flavonoids, and terpenes in the root of *L. martabanica.* The study in TPC revealed that the extracts and their fractions contain various amounts of total phenolics. Therefore, the high antioxidant activity of *L. martabanica* may result from the presence of these compounds. Antioxidant activity study in the plant extract represents an important role. Since substances with low antioxidant activity in vitro will probably show little efficacy in vivo [39]. *L. martabanica *extracts and their fractions exhibited high antioxidant properties. Therefore, we selected crude water extract to investigate the anti-pesticide potential in rats, since this part was utilized for detoxification purposes by people in the highland community.

Chlorpyrifos was used to study the effect of *L. martabanica* on pesticide-exposed rats. The results showed a decrease in AChE activity in the chlorpyrifos-treated group. However, treatment with *L. martabanica* extract tended to restore AChE activity, especially at the high doses of 750 and 250 mg/kg. From this result, *L. martabanica* extract may potentially be used as an anti-pesticide agent. Organophosphate pesticide toxicity is mainly due to AChE inhibition, which causes acetylcholine accumulation. Other mechanisms are involved in oxidative stress and free radical generation [40,41]. Oxidative stress induction by pesticides may occur in many ways [6]. The central mechanism results from the autoxidation process, which increases reactive oxygen species (ROS) production. Some pesticides can alter electron transport chains in mitochondria and endoplasmic reticulum, leading to ROS overproduction. Moreover, pesticides can also inhibit antioxidant and associated enzymes or inhibit the biosynthesis of antioxidants such as glutathione [6]. It has been reported that antioxidant enzymes, such as superoxide dismutase (SOD), catalase, and glutathione-S-transferase activities, are decreased in chlorpyrifos intoxication [42].

The flavonoids found in many plants are powerful natural substances to scavenge free radicals [36,38]. The antioxidant activity of flavonoids is reported to correlate with polyphenolic structures [43,44]. In our study, the crude water extract consists of phenolic compounds and flavonoids. Therefore, high antioxidant activities may relate to their structure. We suggest that the anti-pesticide potential of *L. martabanica* extract may be partly due to antioxidant properties. It has been reported that acute oral poisoning by chlorpyrifos involves AChE inhibition [45]. In this study, chlorpyrifos-exposed rats showed signs of toxicity, such as piloerection, irregular respiratory patterns, and isolation from the group. The rats treated with *L. martabanica* extract exhibited behavior similar to the normal group. This effect may result from the anti-pesticide potential of *L. martabanica* extract, which could increase AChE activity.

The hematological analysis is the sensitive indicator to assess the toxicity of the plant extract, since the ingestion of toxic compounds could change various parameters in the hematological system [46]. RBC, white blood cell (WBC), and platelets are used to assess the health of laboratory animals. RBC function involves carrying oxygen from the lungs to the body as well as bringing carbon dioxide back to the lungs. RBCs volume decreased in various conditions, including blood loss, immune-mediated hemolysis, inflammatory disease, renal disease, iron deficiency, myelodysplastic disease, genetic disorders, and neoplasia [47]. WBCs serve to eliminate foreign bodies. Therefore, WBC counts may be used to indicate infection, inflammation, and immune system disorders. PLTs are cells that play an important role in blood clotting. A decreased number of platelets or thrombocytopenia is associated with bleeding [47]. The MCV, MCH, and MCHC represent the size of RBC, the amount of hemoglobin in RBC, and the concentration of hemoglobin in an average RBC, respectively [48]. In this study, the values of MCV, MCH, and MCHC increased in the pesticide-exposed rats. This effect may be compensated by increasing the RBC volume to carry the oxygen supply to the rats’ bodies. Our results correlate with a previous study, in which short-term chlorpyrifos exposure caused a decrease of RBCs and HCT and an increase of MCV, MCH, and MCHC in rats [49,50] However, abnormal hematological parameters could be normalized by treatment with *L. martabanica* extract.

Blood chemistry profiles were measured to assess the physiological and pathology state of vital organs of laboratory animals. The kidney function of rats was assessed by BUN and Cr measurement. The results showed a slight increase in BUN and Cr levels in chlorpyrifos-exposed rats. *L. martabanica* treatment tended to reduce BUN and Cr levels, and these values did not differ from the normal group. Release of AST, ALT, and ALP into the blood occurred after liver cells injury. Therefore, the increase in these enzymes indicated liver cell damage. Bilirubin (TB and DB) is an indication of the detoxification function of the liver. An increase in TB and DB in the bloodstream demonstrated abnormal liver function in the detoxification of toxic substances. The levels of TP and ALB reflect the synthetic function of the liver. In chronic liver diseases, the liver loses the synthetic function, which causes a reduction in TP and ALB levels in the bloodstream. ALB also decreased in renal diseases due to loss from the glomerulus [51,52]. Chlorpyrifos-induced toxicity in rats involved in free radical generation leading to liver cell damage and increased liver enzymes [40,41]. In this study, the liver biomarkers TB, DB, AST, ALT, and ALP increased in chlorpyrifos-exposed rats. These results correlate with previous research [40,41,53]. *L. martabanica* extract treatment may protect the liver cell, as the results revealed the reduction of abnormal values of liver biomarkers. All clinical chemistry values of rats treated with the extracts comparable with those of the normal group. The results showed no aberration to indicate the presence of a physiological abnormality in the rats and the pathology of the vital organs such as the liver and kidneys. There was little difference in elevation or decline in some clinical chemistry values, which were not affected in liver and kidney function, and all values were within the reference ranges [28].

The liver histology results revealed the necrosis of hepatic cells in the chlorpyrifos-exposed group with an increasing number of sinusoids dilatation. The results correlate with those previously reported by Albasher and colleagues [54]. *L. martabanica *extract treatment helped to protect the liver cells from damage in the rats. The histopathology results showed a reduced number of sinusoid dilation and no hepatic necrosis in the extract-treated group. The liver cells of rats in the treatment group varied in shapes and sizes and exhibited vesicles with small nuclei. These are signs of hepatic regeneration that cause the restoration of the total number and mass of hepatocytes. Loss of liver mass can be induced by toxic chemicals administration. This process is followed by an inflammatory response and a regeneration response [55]. We suggest that the *L. martabanica* extract may improve liver function and protect against oxidative damage induced by chlorpyrifos. 

## 4. Materials and Methods

### 4.1. Plant Material

*Litsea martabanica* was collected from Chiang Mai province, Thailand. The plant material was identified by the taxonomist. The voucher specimen was deposited in the Queen Sirikit Botanical Garden (No. WP 7185). The roots of *L. martabanica* were selected, reduced in size and dried in the hot air oven until the moisture was less than 10%, after which they were pulverized. The powder of the plant material was evaluated for their quality of raw material following the methods described in the Thai Herbal Pharmacopoeia 2018 [27].

### 4.2. Extraction of L. martabanica (Root)

The extraction process followed traditional methods. The coarse powder of the roots was extracted by decoction using water as a solvent. The extract was filtrated, concentrated until % total soluble solid or Brix = 3, and then dried by a spray dryer. Besides the water extract, the root of *L. martabanica* was extracted with 95% ethanol. The crude ethanol extract was separated by partition technique using n-hexane and chloroform (CHCl_3_), respectively. The fractions of n-hexane, CHCl_3_, and aqueous ethanol were evaporated and used for in vitro antioxidant activity study.

### 4.3. Chemical Profile by High Performance Thin Layer Chromatography

The extract samples (1 mg) were separately dissolved in 1 mL of aqueous ethanol as a test solution. Standards (apigenin, caffeic acid, gallic acid, kaemferol, pinene, and quercetin) were each prepared in the concentration of 1 mg/1 mL. A CAMAG (Muttenz, Switzerland) HPTLC system, comprising a Linomat 5 automatic applicator with a 10 mL syringe, CAMAG automatic developing Chamber 2 (ADC 2), Camag TLC scanner 4, and winCATS software version 1.4 was used. For HPTLC fingerprinting analysis, 2 μL of the test solution and 2 μL of the standard solution were loaded as 8 mm band length in the Silica Gel GF_254_ TLC plate. The plate was kept in TLC twin trough developing chamber (after saturated with solvent vapor) with the mobile phase (Ethyl acetate: Methanol:Water = 70:26:4). (Ethyl acetate: Methanol:Water = 70:26:4). Densitometric scanning was performed with a TLC scanner equipped with winCATS software. The plate was scanned at 254, 280, and 320 nm. The plate was kept in a photo-documentation chamber (CAMAG TLC Visualizer 2) and captured the images at White light, UV 254 nm, and UV 366 nm. The developed plate was sprayed with respective spray reagents (anisaldehyde, DPPH, natural product spraying reagent) and dried at 100 °C in a hot air oven.

### 4.4. DPPH Assay

The DPPH assay was performed to evaluate the free radical scavenging activity of the extract fractions. The DPPH (2,2-diphenyl-1-picrylhydrazyl) was dissolved in methanol at a final concentration of 80 µg/mL [56]. The extracts were diluted in various concentrations. The assay method was done on a 96-wells plate as described by Phull and co-workers [57]. Each diluted extract (20 µL) was pipetted into a separate well. Then, DPPH solution (180 μL) was added and mixed. The plate was incubated at room temperature for 30 min in the dark. The absorbance was measured at 517 nm using a microplate reader. Gallic acid and methanol were used as a reference standard and control, respectively. The percentage of DPPH scavenging activity was calculated using the formula as Equation (1):(1)$$\mathrm{DPPH}\;\mathrm{scavenging}\;\left(\%\right)=\frac{\mathrm{Absorbance}\;\mathrm{of}\;\mathrm{control}\;-\;\mathrm{Absorbance}\;\mathrm{of}\;\mathrm{test}\;\mathrm{sample}}{\;\mathrm{Absorbance}\;\mathrm{of}\;\mathrm{control}}\times 100$$

The concentration of the sample required for the inhibition of 50% of DPPH radicals was expressed as IC_50_ values [56]. The IC_50_ values were calculated using linear regression analysis and used to indicate the antioxidant capacity of the extract.

### 4.5. Superoxide Radical Assay

The assay was performed to assess the antioxidant activity of the test sample in scavenging superoxide free radicals. Phenazine methosulfate (PMS) and nicotinamide adenine dinucleotide (NADH) were used to generate superoxide free radicals in the system. Then, superoxide radicals reduced nitro blue tetrazolium (NBT) to purple formazan [58,59]. The reagents PMS (25 μM), NADH (0.5 mM), and NBT (0.2 mM) were dissolved in phosphate buffer solution (pH 7.4). To perform the assay, NBT solution (50 μL), NADH solution (50 μL), and different concentrations of samples (50 μL) were pipetted into a 96-well plate and mixed. Then, PMS solution (50 μL) was added to the well. The plate was mixed and sat at room temperature for 10 min. Then measured the OD at 560 nm using a microplate reader. Gallic acid and phosphate buffer solution were used as a reference standard and control, respectively. The percentage of superoxide radicals scavenging and the IC_50_ values were calculated by the same equation as the DPPH assay.

### 4.6. 2,2′-Azino-Bis-(3-Ethylbenzothiazoline-6-Sulfonic Acid) (ABTS) Assay

ABTS radical scavenging activity of the extracts was conducted according to method described by Sharopov and co-workers [60]. The ABTS reagent was prepared by dissolving 38 mg ABTS reagent in 10 mL deionized purified water (final concentration was 7.0 mM). Then, 6.5 mg potassium persulfate was added to the ABTS solution and allowed to react for 16 h to form the stable ABTS^•+^ radical cation. After 16 h of incubation, ABTS solution was diluted with distilled water to obtain a final absorbance value between 0.700 ± 0.02 at 630 nm. To perform the ABTS assay, 10 µL of diluted extracts were loaded into a 96-well plate, and 190 µL of ABTS reagent was added to the well. The absorbance was measured at 630 nm after 15 min of mixture reaction. Trolox was used as standard substance. The results were expressed in milligram equivalents of trolox per gram of dry weight extract.

### 4.7. Ferric Reducing Antioxidant Power (FRAP) Assay

The FRAP assay was conducted according to the FRAP assay method with slight modifications [61,62]. FRAP reagent was prepared freshly by mixing 300 mM acetate buffer pH 3.6, 10 mM TPTZ (2,4,6-tri(2-pyridyl)-s-triazine) in 40 mM HCl, and 20 mM FeCl_3_·6H_2_O in a volume ratio 10:1:1. The FRAP working solution was warmed at 37 °C for 30 min prior to the assay. For the determination of the FRAP assay, 10 μL of the diluted test compound was mixed with 190 μL FRAP reagent in a 96-well plate, left for 5 min at room temperature, and the absorbance was measured at 595 nm in a microplate reader [60]. Ferrous sulphate (FeSO_4_) was used to generate the standard curve. FRAP values were expressed as mM Fe (II)/g dry weight extract.

### 4.8. Total Phenolics Content

Total phenolics content of the extracts was determined using Folin-Ciocalteu method [63] with slight modifications. The test sample (10 μL) of extract diluted appropriately in dimethyl sulfoxide (DMSO) was mixed with 100 μL Folin-Ciocalteu’s phenol reagent freshly diluted 1/10 with distilled water. After five minutes of incubation, 100 μL of 7.5% Na_2_CO_3_ solution was added, and left for 60 min, before measurement of absorbance at 650 nm in a microplate reader. Appropriate blanks (DMSO) and standard (gallic acid in DMSO) were run simultaneously. The phenolic content was calculated as gallic acid equivalents (GAE mg/g dry weight extract) on the basis of a standard curve of gallic acid [64].

### 4.9. Anti-Pesticide Potential

#### 4.9.1. Animals

Male Sprague-Dawley rats, weighing 180–200 g, were detained from the National Laboratory Animal Center, Nakorn Pathom. They were housed under standard environmental conditions of temperature at 24 ± 1 °C under a 12 h dark-light cycle. All animals had free access to drinking water and standard pellet diet (082 C.P. MICE FEED, S.W.T. Co., Ltd., Samut Prakan, Thailand). They were acclimatized at least one week before starting the experiments. The Animal Ethics Committee of Faculty of Medicine, Chiang Mai University approved all experimental protocols, No. 49/2559.

#### 4.9.2. Experimental Groups

The anti-pesticide potential of *L. martabanica *water extract was modified from the method previously reported [65]. Male rats were divided into five groups of six animals each.

Group 1, normal group: rats received no treatment, only 2 mL/kg of distilled water by gavage daily for 16 days and were used to determine the normal values of tested parameters.

Group 2, control group: rats received 2 mL/kg of distilled water by gavage daily for 16 days (four rounds).

Group 3, test group: rats received the cycle dose of the root water extract of *L. martabanica* 7.5 mg/kg for 2 days, then 2.5 mg/kg for 2 days; each rat received the extract daily for 16 days (four rounds).

Group 4, test group: rats received the cycle dose of the root water extract of *L. martabanica* 75 mg/kg for 2 days, then 25 mg/kg for 2 days; each rat received the extract daily for 16 days (four rounds).

Group 5, test group: rats received the cycle dose of the root water extract of *L. martabanica* 750 mg/kg for 2 days, then 250 mg/kg for 2 days; each rat received the extract daily for 16 days (four rounds).

The rats in group 3 to 5 received the extract in a way that mimics the traditional methods of tribal communities on the highlands. Distilled water and *L. martabanica* extract were orally given to the rats 30 min prior to receiving chlorpyrifos (Sigma) at a dose of 16 mg/kg. On the 17th day, all rats were anesthetized with phenobarbital sodium (50 mg/kg, intraperitoneally). A cannula was inserted into the common carotid artery for blood collection. A blood sample of each rat was distributed into a clean tube without anticoagulant and a tube with anticoagulant (EDTA).

### 4.10. Assay of AChE Activity

AChE activity was determined by using an AChE assay kit according to the assay protocols (Sigma) [66]. Briefly, whole blood samples were diluted (1:40) with assay buffer, pH 7.5. Then, 10 µL of samples was transferred into separate wells of the 96-well plate and 190 mL of the working reagent were added to all samples. The reaction mixtures were mixed and incubated at room temperature. The absorbance was monitored at 2 min and 10 min, respectively, by a microplate reader at 412 nm. AChE activity was calculated using the formula as Equation (2):(2)$$\mathrm{AChE}\;\mathrm{activity}\;\left(\mathrm{units}/L\right)=\frac{A\;\left(\mathrm{sample}\right)\;\mathrm{at}\;10\;\min-\;A\;\left(\mathrm{sample}\right)\;\mathrm{at}\;2\;\min}{\;A\;\left(\mathrm{calibrator}\right)\;\mathrm{at}\;10\;\min-\;A\;\left(\mathrm{blank}\right)\;\mathrm{at}\;2\;\min}\times n\times 200$$
A = absorbance; 200 = equivalent activity (units/L) of the calibrator when assayed is read at 2 and 10 min; n = dilution factor

### 4.11. Observation of Behavioral Change and Toxicological Signs

Behavior change after chlorpyrifos and *L. martabanica* extracts administration were observed in the rats. The signs of toxicity, such as piloerection, diarrhea, tremor, lack of coordination, salivation, lacrimation, and others, were observed and recorded [65].

### 4.12. Body Weight Change, Internal Organ Weight, and Histopathological Studies

During the experiment, the rats’ body weight was measured once daily. On day 17, the rats were sacrificed and the liver removed for weighing and gross pathological detection. The liver was preserved in 10% neutral buffered formaldehyde solution for histopathological examination.

### 4.13. Hematology Analysis

Blood samples were collected and determined, and blood count was completed using the automatic hematology system to evaluate red blood cell (RBC), white blood cell (WBC), hemoglobin (HGB), hematocrit (HCT), mean corpuscular volume (MCV), mean corpuscular hemoglobin (MCH), mean corpuscular hemoglobin concentration (MCHC), platelet (PLT), neutrophil (Nu), lymphocyte (lymph), monocyte (Mono), eosinophil (E), and basophil (Ba).

### 4.14. Blood Chemistry Analysis

Clotted blood samples were centrifuged to collect the serum. Blood chemistry, such as blood urea nitrogen (BUN), creatinine (Cr), total protein (TP), albumin (ALB), total bilirubin (TB), direct bilirubin (DB), aspartate aminotransferase (AST), alanine aminotransferase (ALT), and alkaline phosphatase (ALP), was analyzed.

### 4.15. Statistical Analysis

For in vitro antioxidant assays, data were presented as the mean ± standard error of the mean (S.E.M) from three independent experiments. For in vivo experiments, statistical comparisons between the mean of each group were analyzed using the one-way ANOVA with Post Hoc multiple comparison. A value of *p* < 0.05 was considered statistically significant.

## 5. Conclusions

From our results, it can be concluded that *L. martabanica* extract possesses anti-pesticide potential, which may be partly from antioxidant properties. This study provides scientific data to support the use of *L. martabanica* as folkloric medicines. However, the other pharmacological activities and underlying mechanisms should be studied.

## Figures and Tables

**Figure 1 molecules-26-01906-f001:**
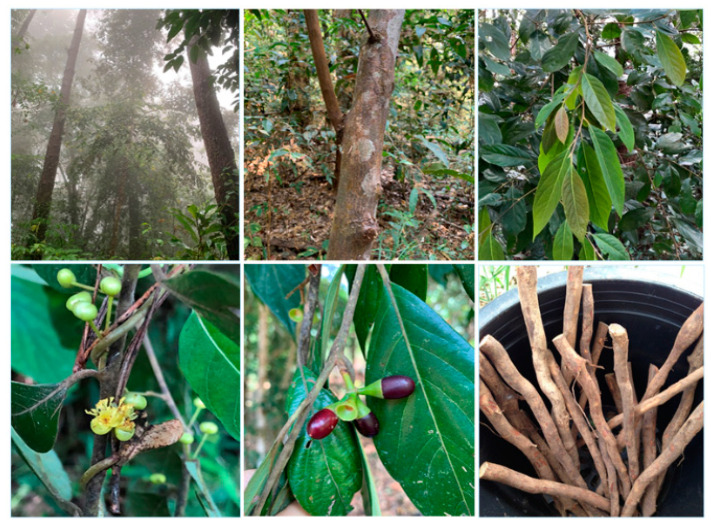
*Litsea martabanica* (Kurz) Hook.f.

**Figure 2 molecules-26-01906-f002:**
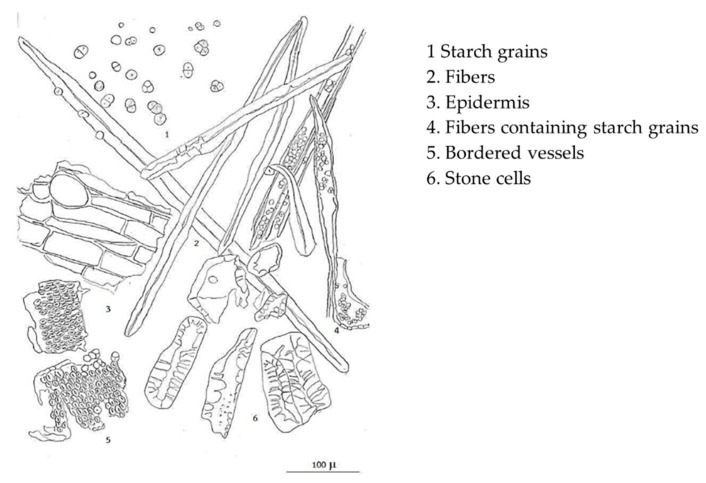
Microscopic character of powder of *L. martabanica* (*root*).

**Figure 3 molecules-26-01906-f003:**
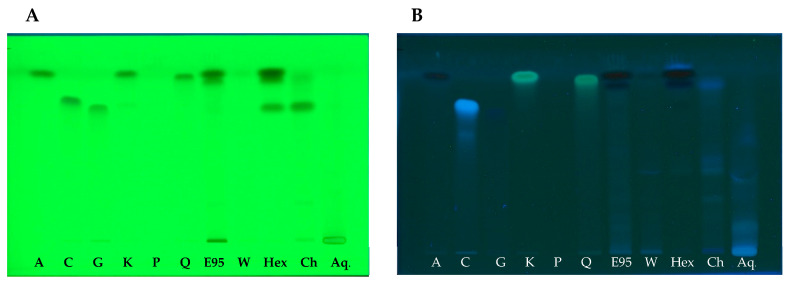
The high-performance thin-layer chromatography (HPTLC) chromatogram (**A**) at 254 nm; (**B**) at 366 nm. A = apigenin; C = caffeic acid; G = gallic acid; K = kaemferol; P = pinene; Q = quercetin; E = ethanolic crude extract; W = water crude extract; Hex = n-hexane fraction; Ch = chloroform fraction; and Aq = aqueous alcohol fraction.

**Figure 4 molecules-26-01906-f004:**
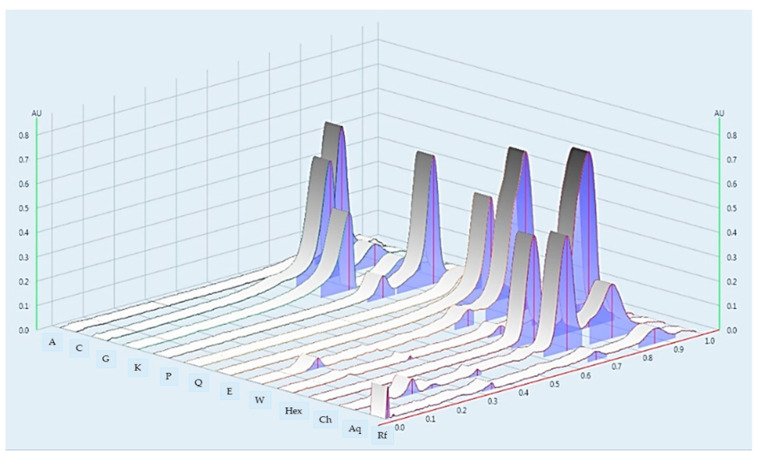
HPTLC Chromatogram at 254 nm. A = apigenin; C = caffeic acid; G = gallic acid; K = kaemferol; P = pinene; Q = quercetin; E = ethanolic crude extract; W = water crude extract; Hex = n-hexane fraction; Ch = chloroform fraction; and Aq = aqueous alcohol fraction. Rf, rate of flow; AU, absorbance units.

**Figure 5 molecules-26-01906-f005:**
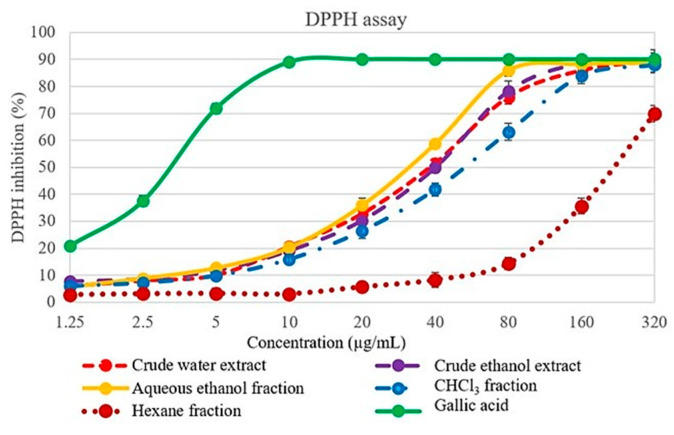
Effect of *L. martabanica* fractions on 2,2′-diphenyl-1-picrylhydrazyl (DPPH) free radicals scavenging. Values are expressed as mean ±S.E.M. from three independent experiments.

**Figure 6 molecules-26-01906-f006:**
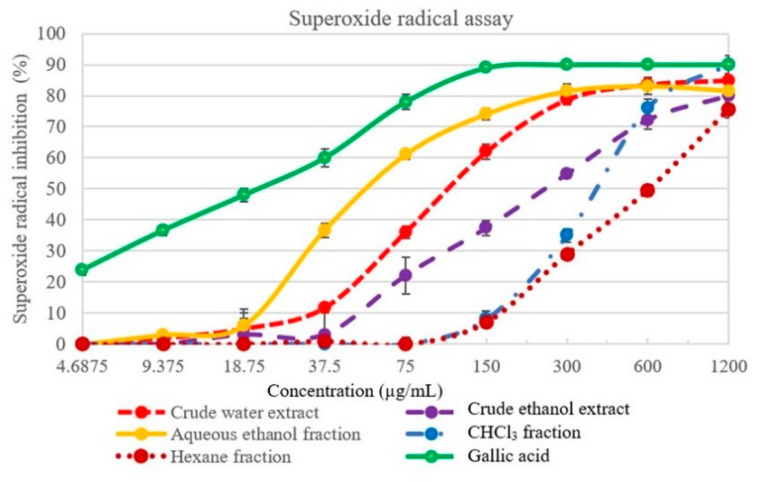
Effect of *L. martabanica* fractions on superoxide radical scavenging. Values are expressed as mean ±S.E.M. from three independent experiments.

**Figure 7 molecules-26-01906-f007:**
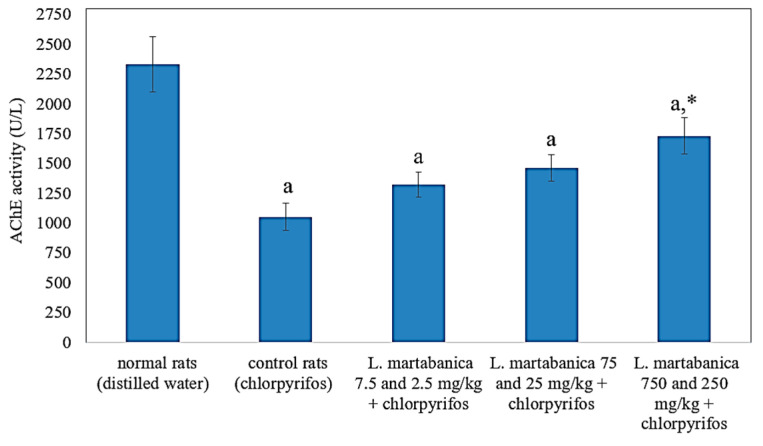
Effect of *L. martabanica* water extract on AChE activity. Values are expressed as mean ± S.E.M. ^a^ Significantly different from the normal rats, *p* < 0.05. * Significantly different from the control rats, *p* < 0.05, according to one-way ANOVA.

**Figure 8 molecules-26-01906-f008:**
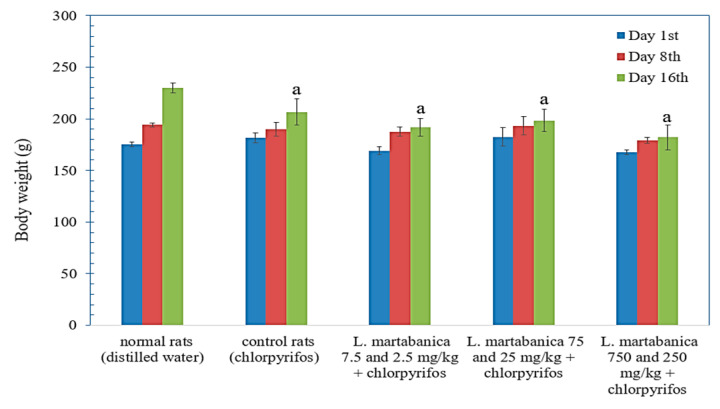
The effect of *L. martabanica* water extract on the body weight of rats. Values are expressed as mean ±S.E.M. (*n* = 6). ^a^ Significantly different from normal rats (*p* < 0.05), according to one-way ANOVA.

**Figure 9 molecules-26-01906-f009:**
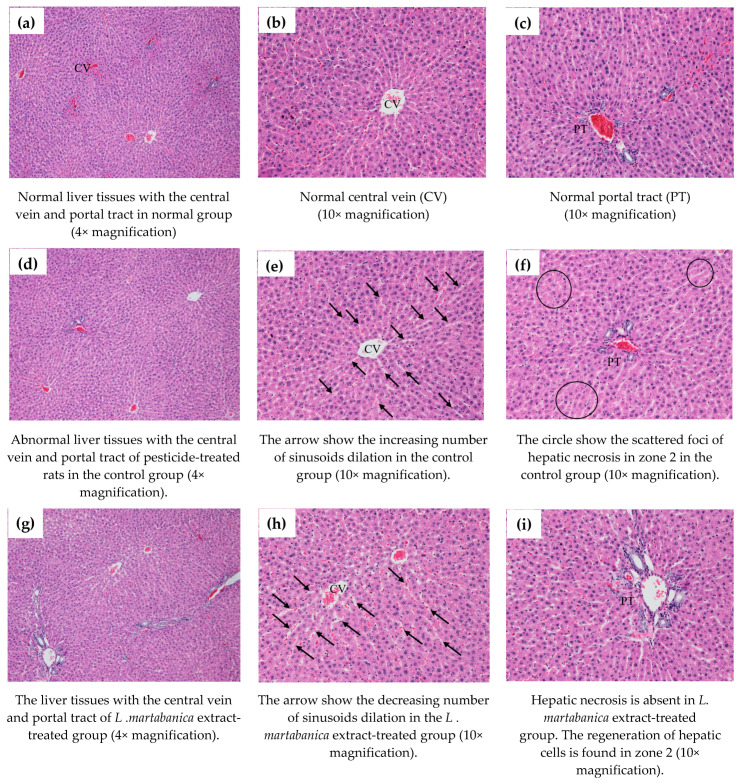
Histopathologic results of rat liver. (**a**–**c**): Histology results of the normal rat liver in the normal group. (**d**–**f**): Histology results of the rat liver in the control group receiving chlorpyrifos. The picture shows the dilation of sinusoids (arrow) and the scattered foci of hepatic necrosis in zone 2 (circle). (**g**–**i**): Histology results of the group that received high doses of *L. martabanica* extract (750 and 250 mg/kg). The number of sinusoids dilation is decreased.

**Table 1 molecules-26-01906-t001:** Physical and chemical properties of root of *L. martabanica*.

Test	Result
Foreign matter (%*w/w*)	Not found
Ethanol extractive content (%*w/w*)	33.0534 ± 0.06
Water extractive content (%*w/w*)	24.1200 ± 0.04
Loss on drying (%*v/w*)	9.6867 ± 0.02
Total ash (%*w/w*)	1.4645 ± 0.01
Acid-insoluble ash (%*w/w*)	0.1578 ± 0.00
Chemical composition	phenolics, flavonoids, terpenes

Values are expressed as mean ± standard error of the mean (S.E.M.) from three independence experiments.

**Table 2 molecules-26-01906-t002:** The IC_50_ values of *L. martabanica* root extracts in DPPH assay and superoxide radical assay.

Test Samples	IC_50_ (µg/mL)	
	DPPH	Superoxide Radical Scavenging
Gallic acid	2.7 ± 0.01	23.8 ± 3.9
Crude water extract	42.8 ± 4.1	118.6 ± 10.4
Crude ethanol extract	44.2 ± 2.3	259.3 ± 28.9
Hexane fraction	233.8 ± 21.7	593.5 ± 9.7
CHCl_3_ fraction	57.0 ± 1.6	417.7 ± 10.1
Aqueous ethanol fraction	32.4 ± 1.5	58.9 ± 5.2

Values are expressed as mean ± S.E.M. from three independent experiments. IC_50_, the half maximal inhibitory concentration; DPPH, 2,2′-diphenyl-1-picrylhydrazyl.

**Table 3 molecules-26-01906-t003:** Antioxidant properties of *L. martabanica* root extracts in ABTS assay, FRAP assay, and TPC assay.

Test Samples	ABTS	FRAP	TPC
	(TE mg/g Extract)	(mM Fe (II)/g Extract)	GAE (mg/g Extract)
Crude water extract	78.2 ± 1.4	368.9 ± 23.4	42.2 ± 4.5
Crude ethanol extract	188.0 ± 0.9	1376.2 ± 60.4	147.9 ± 2.8
Hexane fraction	98.8 ± 10.9	275.4 ± 39.5	56.5 ± 4.9
CHCl_3_ fraction	163.4 ± 8.8	1554.1 ± 23.9	173.1 ± 0.4
Aqueous ethanol fraction	71.1 ± 10.6	418.6 ± 31.0	39.3 ± 5.1

Values are expressed as mean ± S.E.M. from three independent experiments. ABTS, 2,2′-azino-bis-(3-ethylbenzothiazoline-6-sulfonic acid); FRAP, ferric reducing antioxidant power; TPC, total phenolic content; TE, trolox equivalent; GAE, gallic acid equivalent.

**Table 4 molecules-26-01906-t004:** Effect of *L. martabanica* water extract on the liver weight of rats.

Group	Liver Weight (g)
Normal rats	10.7 ± 0.58
Control rats (chlorpyrifos)	11.89 ± 0.65
*L. martabanica* extract + chlopyrifos	
7.5 and 2.5 mg/kg	12.37 ± 0.96
75 and 25 mg/kg	9.47 ± 0.90 *
750 and 250 mg/kg	11.21 ± 0.70

Values are expressed as mean ±S.E.M. (*n* = 6). * Significantly different from control rats (*p* < 0.05).

**Table 5 molecules-26-01906-t005:** Effect of *L. martabanica* water extract on the hematological values of rats.

Parameters	Normal Rats	Control Rats	*L. martabanica.* Water Extract (mg/kg)		
			7.5 and 2.5	75 and 25	750 and 250
RBC (×10^6^/µL)	7.0 ± 0.13	5.8 ± 0.80	7.5 ± 2.45	7.0 ± 0.22	6.9 ± 0.23
HGB (g/dL)	13.6 ± 0.22	12.5 ± 0.41	11.5 ± 1.02	13.7 ± 0.41	13.4 ± 0.43
HCT (%)	41.3 ± 0.87	36.5 ± 3.88	38.6 ± 0.64	41.7 ± 1.62	41.3 ± 1.20
MCV (fL)	59.4 ± 0.28	64.3 ± 2.92 ^a^	59.2 ± 0.44 *	59.2 ± 0.56 *	59.2 ± 0.43 *
MCH (pg)	19.6 ± 0.12	25.8 ± 4.20 ^a^	19.3 ± 0.13 *	19.5 ± 0.11 *	19.2 ± 0.18 *
MCHC (g/dL)	33.0 ± 0.17	39.2 ± 4.38 ^a^	32.7 ± 0.22 *	32.9 ± 0.33 *	32.4 ± 0.19 *
PLT (×10^5^/µL)	7.53 ± 0.34	8.78 ± 1.04 ^a^	6.76 ± 0.14 *	7.91 ± 0.50	7.25 ± 0.19
WBC (×10^3^ cells/µL)	2.84 ± 0.49	2.87 ± 0.72	1.68 ± 0.31 ^a,^*	2.72 ± 1.32	2.18 ± 0.37
Nu (cells/µL)	0.39 ± 0.06	0.45 ± 0.21	0.30 ± 0.06	0.31 ± 0.14	0.16 ± 0.05 ^a,^*
Lymph (cells/µL)	2.26 ± 0.38	2.24 ± 0.45	3.00 ± 1.92	2.30 ± 1.12	0.97 ± 0.30 ^a,^*
Mono (cells/µL)	0.15 ± 0.04	0.15 ± 0.05	0.09 ± 0.02	0.09 ± 0.06	0.05 ± 0.02
E (cells/µL)	0.00 ± 0.01	0.00 ± 0.02	0.00 ± 0.00	0.00 ± 0.01	0.00 ± 0.00
Ba (cells/µL)	0.00 ± 0.00	0.00 ± 0.00	0.00 ± 0.00	0.00 ± 0.00	0.00 ± 0.00

Values are expressed as mean ±S.E.M. (*n* = 6). ^a^ Significantly different from the normal rats (*p* < 0.05), * Significantly different from the control rats (chlorpyrifos) (*p* < 0.05), according to one-way ANOVA. RBC, red blood cell; HGB, hemoglobin; HCT, hematocrit; MCV, mean corpuscular volume; MCH, mean corpuscular hemoglobin; MCHC, mean corpuscular hemoglobin concentration; PLT, platelet; WBC, white blood cell; Nu, neutrophil; Lymph, lymphocyte; Mono, monocyte; E, eosinophil; Ba, basophil.

**Table 6 molecules-26-01906-t006:** Effect of *L. martabanica* water extract on the blood chemistry values of rats.

Parameters	Normal Rats	Control Rats	*L. martabanica.* Water Extract (mg/kg)		
			7.5 and 2.5	75 and 25	750 and 250
BUN (mg/dL)	18.4 ± 0.74	20.1 ± 1.02	15.6 ± 0.91 *	16.6 ± 1.47	19.9 ± 2.38
Cr (mg/dL)	0.66 ± 0.01	0.79 ± 0.02	0.67 ± 0.01	0.57 ± 0.10 *	0.72 ± 0.04
TP (g/dL)	5.9 ± 0.07	6.9 ± 0.45 ^a^	5.8 ± 0.11 *	6.1 ± 0.23	6.1 ± 0.40
ALB (g/dL)	2.9 ± 0.04	3.4 ± 0.24 ^a^	2.9 ± 0.02 *	3.1 ± 0.14	3.0 ± 0.19
TB (mg/dL)	0.09 ± 0.01	0.16 ± 0.03 ^a^	0.10 ± 0.01 *	0.10 ± 0.02 *	0.08 ± 0.02 *
DB (mg/dL)	0.03 ± 0.01	0.09 ± 0.01 ^a^	0.03 ± 0.00 *	0.05 ± 0.00 *	0.04 ± 0.01 *
AST (U/L)	85 ± 5.57	133 ± 9.20 ^a^	85 ± 4.24 *	92 ± 9.16 *	92 ± 6.70 *
ALT (U/L)	30 ± 7.65	57 ± 5.26 ^a^	45 ± 3.35 *	41 ± 5.33 *	38 ± 2.55 *
ALP (U/L)	165 ± 9.26	214 ± 19.69 ^a^	180 ± 11.99 *	137 ± 10.46 *	174 ± 14.86 *

Values are expressed as mean ±S.E.M. (*n* = 6). ^a^ Significantly different from the normal rats (distilled water) (*p* < 0.05), * Significantly different from the control rats (distilled water + chlorpyrifos), (*p* < 0.05). BUN, blood urea nitrogen; Cr, creatinine; TP, total protein; ALB, albumin, TB, total bilirubin; DB, direct bilirubin; AST, aspartate aminotransferase; ALT, alanine aminotransferase; ALP, alkaline phosphatase.

## Data Availability

The data presented in this study are available in this article.

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
