# Peer review of "In Vitro Antioxidant Activity of *Litsea martabanica* Root Extract and Its Hepatoprotective Effect on Chlorpyrifos-Induced Toxicity in Rats"

_molecules, 2021, doi:10.3390/molecules26071906_

Round 1

Reviewer 1 Report

The manuscript entitled “In vitro Antioxidant Activity of Litsea martabanica Root Extract and Its hepatoprotective Effect on Chlorpyrifos-Induced Toxicity in Rats”, authored by Phraepakaporn Kunnaja and colleagues, deals with the investigation of the antioxidant properties of Litsea martabanica root extract. In particular, the extracts was evaluated via DPPH and superoxide radicals scavenging assay. Moreover, the authors investigated the potential use of the extract as a detoxifying agent, evaluating this action in in vivo model system exposed to chlorpyrifos. The link between the antioxidant action and the anti-pesticide potential of the extract, maybe, is too strong. However, the study and the purpose is really alternative and interesting. In particular, the histological images from animals exposed, or not, to the different treatments is very interesting.

The introduction is really well written. It follows a very logical thread, and reports all the information necessary to motivate the study. However, in the manuscript there is a strong lack in statistical methods, and in the tests used to highlight the differences between the treatments and the respective controls. The authors should re-perform the statistical analyses following the indications given in this report. Moreover, the section related to phytochemical investigation could be more studied through the execution of simple spectrophotometric assays.

MAJOR:

  • Data contained in Table 2 should be subjected to one-way ANOVA statistical test. In particular, this test should be performed within the DPPH series and within the data obtained by superoxide radical scavenging series, with the aim to identify which extract has the highest value. Once the statistical test has been performed, add different lowercase (for DPPH) or uppercase (for baba) letters. Remember to add the statistical information both in the materials and methods section and in the legend of Table 2
  • The phytochemical investigation of the raw material should be better investigated. The HPTLC analyses are very interesting; however, they do not provide a useful parameter for comparison with other literature data. In particular, the authors should perform Folin-Ciocolteau assay with the aim to quantify the total polyphenol content (TPC). This simple and fast spectrophotometric assay is universally used to compare the content of bioactive compounds within extracts of plant matrices. The comparison of the TPC could be made both within the different types of extracts prepared by the authors, and with those previously reprinted in the literature.
  • Moreover, authors should introduce information the evaluation of antioxidant activity via other spectrophotometric assay. DPPH is an assay well-known aimed to evaluate the antioxidant properties of plant extracts, however this data alone can not be sufficient to demonstrate the real potential of plant extracts. To this purpose, other assays (ABTS, FRAP, ORAC, CuPRAC) are largely used for this purpose. These assays, while measuring all the antioxidant potential of a plant extract, are based on completely different mechanisms of action, which can better elucidate the reasons why an extract shows a better antioxidant potential than another one.
  • TPC, DPPH and data obtained from other assays evaluating antioxidant activity are often correlated with each other. This may be another point of discussion that authors should elaborate on.
  • It would be advisable to perform one-way anova also for the data shown in figure 6 and 7, and reported in Table 4 and

MINOR:

  • Please, report only two significant figures after the comma in the values of Table 1.
  • In Figure 3, authors should replace the number of the sample using the same acronyms of Figure 2 (A, C, G, K, P, etc…)
  • In the caption of both Figure 2 and Figure 3, the different concentration of water and ethanolic extracts should be included
  • Please, modify IC50 -> IC50 all over the main text
  • Please, report two significant figures after the comma in the values of Table 2
  • Please, remove ‘groups’ from the x-axis of the figure 6 and 7.

Author Response

MAJOR:

Point 1: Data contained in Table 2 should be subjected to one-way ANOVA statistical test. In particular, this test should be performed within the DPPH series and within the data obtained by superoxide radical scavenging series, with the aim to identify which extract has the highest value. Once the statistical test has been performed, add different lowercase (for DPPH) or uppercase (for baba) letters. Remember to add the statistical information both in the materials and methods section and in the legend of Table 2

Response 1: The authors thank you for your valuable suggestion to improve this work. We performed DPPH assay and superoxide radical assay as we want to know whether the radical scavenging activity of the plant close to the reference standard. We presented data in Table 2 as the IC50 value in both assays, and we are interested in the low IC50 values of test samples because it’s indicated high anti-oxidant potential. So, the authors think it may not be necessary to show comparative results between the reference standard and the test samples. We have reviewed many anti-oxidant articles and found that they are usually expressed as the IC50 values. Therefore, we would like to express IC50 values of the extracts without comparing them with the others (line 173-174).

Point 2: The phytochemical investigation of the raw material should be better investigated. The HPTLC analyses are very interesting; however, they do not provide a useful parameter for comparison with other literature data. 

Response 2: In this study, HPTLC chromatogram did not indicate the chemical components, but the fingerprinting can be used to analyze the quantification of herbal products, phytochemical and examine adulteration in herbal formulations.

Point 3: The authors should perform Folin-Ciocolteau assay with the aim to quantify the total polyphenol content (TPC). This simple and fast spectrophotometric assay is universally used to compare the content of bioactive compounds within extracts of plant matrices. The comparison of the TPC could be made both within the different types of extracts prepared by the authors, and with those previously reprinted in the literature. Moreover, authors should introduce information the evaluation of antioxidant activity via other spectrophotometric assay. DPPH is an assay well-known aimed to evaluate the antioxidant properties of plant extracts, however this data alone can not be sufficient to demonstrate the real potential of plant extracts. To this purpose, other assays (ABTS, FRAP, ORAC, CuPRAC) are largely used for this purpose. These assays, while measuring all the antioxidant potential of a plant extract, are based on completely different mechanisms of action, which can better elucidate the reasons why an extract shows a better antioxidant potential than another one.

TPC, DPPH and data obtained from other assays evaluating antioxidant activity are often correlated with each other. This may be another point of discussion that authors should elaborate on.

Response 3: The authors agree with the reviewer. We have performed the three assays ABTS, FRAP, and TPC of the different extracts as reviewer suggestions. The results of these assays are presented in text and in Table 3 (Line 175-203). The results from all antioxidant assays (including in DPPH and superoxide radical assay) demonstrated that chloroform fraction possesses high antioxidant property over the other fraction, followed by crude ethanol extract, aqueous ethanol fraction, water extract, and hexane fraction, respectively. However, the water extract is traditionally used as a detoxifying agent by local people in the highland communities. Therefore, we chose this water extract of L. martabanica for in vivo experiments. We added the results of ABTS, FRAP, and TPC in Table 3 and also in the discussion (Line 277-280, 281-282, 284-285, 295-307), materials and methods (Line 401, 409, 442-472), and abstract sections (Line 19-24) in the revised manuscript.

Table 3. Antioxidant properties of L. martabanica root extracts in ABTS assay, FRAP assay, and TPC assay.

Test samples

ABTS

FRAP

TPC

(TE mg/g extract)

(mM Fe (II)/g extract)

GAE (mg/g extract)

Crude water extract

78.2 ± 1.4

368.9 ± 23.4

42.2 ± 4.5

Crude ethanol extract

188.0 ± 0.9

1376.2 ± 60.4

147.9 ± 2.8

Hexane fraction

98.8 ± 10.9

275.4 ± 39.5

56.5 ± 4.9

CHCl3 fraction

163.4 ± 8.8

1554.1 ± 23.9

173.1 ± 0.4

Aqueous ethanol fraction

71.1 ± 10.6

418.6 ± 31.0

39.3 ± 5.1

Values are expressed as mean ± S.E.M. from three independent experiments.

Point 4: It would be advisable to perform one-way anova also for the data shown in figure 6 and 7, and reported in Table 4.

Response 4: Since we have inserted the photo picture of L. martabanica in Figure 1. Therefore, Figure 6 and Figure 7 have changed to Figure 7 and Figure 8, respectively. We also added Table 3 to the revised manuscript, therefore, Table 4 changed to Table 5. Statistical analysis was changed (Line 540-545). We have performed One-way ANOVA for the data in Figure 7 (Line 211-214) and Figure 8 (Line 228-231), and Table 5 as suggested by the reviewer in the revised manuscript (Line 264-266).

MINOR:

Point 1: Please, report only two significant figures after the comma in the values of Table 1.

Response 1: In Table 1 we just want to present the physical and chemical properties of the root of L. martabanica. So, we don’t have a significant number.

Point 2: In Figure 3, authors should replace the number of the sample using the same acronyms of Figure 2 (A, C, G, K, P, etc…)

Response 2: Figure 2 and 3 were changed to Figure 3 and 4. The number of the sample under Figure 4 was replaced by the same acronyms as Figure 3 in the revised manuscript (Line 143-154).

Point 3: In the caption of both Figure 2 and Figure 3, the different concentration of water and ethanolic extracts should be included.

We have performed a new HPTLC chromatogram by using different fractions of L. martabanica (Line 137-140 and 143-154), which were ethanol crude extract, water crude extract, hexane fraction, chloroform fraction, and aqueous ethanol fractions. So, we couldn’t show the different concentrations of water and ethanolic extracts. 

Point 4: Please, modify IC50 -> IC50 all over the main text

Response 4: IC50 was changed to IC50 all over the main text.

Point 5: Please, report two significant figures after the comma in the values of Table 2

Response 5: We would like to express only IC50 values without comparing between groups.

Point 5: Please, remove ‘groups’ from the x-axis of the figure 6 and 7.

Response 5: Figure 6 and 7 were changed to Figure 7 and 8, respectively. We have moved ‘groups’ from the x-axis of the Figure 7 (Line 211-214), and Figure 8 (228-231).

Reviewer 2 Report

This study aimed to evaluate antioxidant activity and anti-pesticide potential of Litsea martabanica root extract. L. martabanica extract exhibited significant effect for detoxifying agent, especially from chlorpyrifos pesticide. In addition, it showed antioxidant properties.

Although there are no data to identify the "molecules" which are responsible for these functions, I agree with the conclusion that this study provides scientific data to support the use of L. martabanica as folkloric medicines.

Thus, I recommend this manuscript for publication in Molecules after minor revisions on the following points.

ABSTRACT
Line 20:  DPPH should be defined.

INTRODUCTION
Line 57: OP should be defined.

FIGURE
Figure 1
Since most readers are not familiar with L. martabanica, it will be nice to include photo images.

TABLE
Table 1.
What "Physical and chemical properties" mean?

The significant number should be considered.

Author Response

The authors thank you very much for your valuable suggestion to improve this work.

Point 1: ABSTRACT
Line 20:  DPPH should be defined.

Response 1: DPPH was defined as 2,2′-diphenyl-1-picrylhydrazyl (Line 20).

Point 2: INTRODUCTION
Line 57: OP should be defined.

Response 2: OP was defined as Organophosphates (Line 59).

Point 3: FIGURE
Figure 1.
Since most readers are not familiar with L. martabanica, it will be nice to include photo images.

Response 3: The photo images of L. martabanica, was added to the revised manuscript as Figure 1 (Line 49, 97-98).

Point 4: TABLE
Table 1.
 4.1 What "Physical and chemical properties" mean?

Response 4.1. Chemical properties mean Ethanol extractive content; Water extractive content; Chemical composition.

Physical properties are foreign matter, loss on drying, total ash, and acid-insoluble ash.

 4.2 The significant number should be considered.

Response 4.2 In Table 1 we just want to present the physical and chemical properties of the root of L. martabanica. So, we don’t have a significant number.

Round 2

Reviewer 1 Report

the authors correctly edited the manuscript as required. Now it can be considered as a potential publication in Molecules.